# MSRA-Net: A Multi-Task Learning Model for Soil Texture Prediction with Dynamic Weighting and Prior Knowledge Soft Constraints

**DOI:** 10.3390/s25216519

**Published:** 2025-10-23

**Authors:** Yun Deng, Yongjian Xu, Yuanyuan Shi

**Affiliations:** 1Guangxi Key Laboratory of Embedded Technology and Intelligent System, Guilin University of Technology, 12 Jiangan Road, Guilin 541004, China; 2002078@glut.edu.cn; 2College of Computer Science and Engineering, Guilin University of Technology, 319 Yanshan Street, Yanshan District, Guilin 541006, China; 3Key Laboratory of Central South Fast-Growing Timber Cultivation of Forestry Ministry of China, Guangxi Forestry Research Institute, No.23 Yongwu Road, Nanning 530010, China; syyfly@163.com

**Keywords:** multi-task learning, soil texture, visible/near-infrared spectroscopy, dynamic weighting, prior knowledge

## Abstract

Accurate and rapid acquisition of soil texture information is crucial to evaluating soil quality, formulating soil and water conservation strategies, and guiding agricultural resource management. Compared with traditional machine learning methods, convolutional neural networks (CNNs) demonstrate superior accuracy in soil texture prediction. To overcome the limitations of existing lightweight models in spectral modeling, such as insufficient single-scale feature representation, limited channel utilization, and branch redundancy, and to meet the demand for lightweight architectures, we propose a novel dynamic feature modeling approach: Multi-scale Routing Attention Network (MSRA-Net). MSRA-Net integrates grouped multi-scale convolutions with an intra-group Efficient Channel Attention (gECA) mechanism, combined with a multi-scale weighting strategy based on a Branch Routing Attention (BRA) mechanism, thereby enhancing inter-channel feature interaction and improving the model’s ability to capture complex spectral patterns. Furthermore, we introduce a multi-task learning variant, MSRA-MT, which employs uncertainty dynamic weighting to balance gradients magnitude across tasks, thereby improving both stability and predictive accuracy. Experimental results on the LUCAS and ICRAF datasets demonstrate that the MSRA-MT model consistently outperforms baseline models in terms of performance and robustness ($RMSE_{mean}$ = 9.190 and $RMSE_{mean}$ = 8.189 for ICRAF and LUCAS, respectively). Prior knowledge-based soft constraints may hinder optimization by amplifying intrinsic noise, rather than improving learning effectiveness.

## 1. Introduction

Soil texture is one of the key indicators characterizing physical (mechanical) properties [1]. To a certain extent, soil texture also affects the chemical and biological properties of soils [2,3]. The ability to rapidly and accurately obtain soil texture information is of great practical significance for evaluating soil quality, formulating soil and water conservation strategies, and guiding agricultural resource management [4]. Moreover, soil texture serves as a critical indicator for assessing land degradation and wind erosion susceptibility [5,6]. It has an important reference value for identifying potential degradation areas and developing differentiated governance measures [7]. Against the backdrop of challenges to agricultural sustainability, a series of large-scale soil data platforms [8,9], have incorporated high-resolution soil texture monitoring into their research scope [10], providing sufficient conditions for validating spectral modeling methods.

Soil texture (clay, silt, and sand) is typically determined using laser particle analyzers or sedimentation methods [11,12]. Although these conventional approaches offer high accuracy, they are unsuitable for the large-scale analyses required in precision agriculture [13]. In contrast, visible/near-infrared (Vis–NIR) diffuse reflectance spectroscopy offers advantages such as rapid analysis, non-destructive measurement, low cost, and suitability for large-scale applications. However, several challenges remain in practical applications. For instance, the spectral response to soil texture is indirect and relatively weak [14]. Conventional methods struggle to capture the complex nonlinear relationships between soil texture and high-dimensional spectral data, resulting in limited predictive accuracy [15].

Over the past two decades, researchers have enhanced and filtered spectral band features using preprocessing methods such as Multiplicative Scatter Correction (MSC) [16,17], Derivative Transformations (DTs) [18], or feature selection methods such as Competitively Adaptive Reweighted Sampling (CARS) [19,20]. When combined with statistical learning and machine learning methods, such as PLSR, RF, and ensemble learning methods like XGBoost [21], stable predictive performance is demonstrated [22]. However, their predictive capability remains limited when confronted with complex, high-dimensional, nonlinear relationships [23]. Additionally, the limited transferability of such models is also hindering their application at large spatial scales.

In contrast, deep learning can automatically learn multi-level and multi-scale spectral features, demonstrating strong capability in modeling large datasets and complex nonlinear relationships [24,25]. Some researchers have adopted Inception-like multi-branch convolutional architectures to enable parallel extraction of multi-scale features [26]. Other researchers have employed Temporal Convolutional Networks (TCNs) based on causal and dilated convolutions aiming to expand the receptive field while preserving sequential dependencies [27]. Similarly, sequence-dependent modeling, Recurrent Neural Networks (RNNs), and their variants, such as Long Short-Term Memory (LSTM), rely on gating mechanisms to model long-term dependencies [28,29]. In addition, the Transformer architecture has demonstrated outstanding performance in modeling long-range dependencies and full-spectrum contextual relationships, owing to its global self-attention mechanism [30]. These methods have demonstrated superior accuracy in soil texture prediction compared with traditional machine learning approaches.

However, most researchers have employed a single-task modeling strategy, in which the contents of clay, silt, and sand are predicted independently. Such approaches fail to fully exploit the intrinsic constraints and correlations among the three components, which often leads to inconsistent prediction results [31]. In contrast, multi-task learning (MTL) employs a compact shared model to simultaneously learn multiple related tasks [32,33], thereby facilitating knowledge transfer and enhancing overall generalization capacity [34,35]. This approach enables better capture of the coupled relationships among soil texture components and improves the model’s generalization performance. However, multi-task learning methods are prone to the “negative transfer” problem [36,37], and the model struggles to achieve satisfactory accuracy across all tasks due to the lack of an effective task weight balancing mechanism [38].

To address the limitations of traditional convolutional neural networks in spectral modeling, including insufficient single-scale feature expression, limited channel utilization, and branch redundancy, as well as the need for lightweight models, a novel dynamic feature modeling method is proposed. This method combines multi-scale convolution and Efficient Channel Attention (ECA) [39] for predicting soil texture.

The primary work of this paper includes the following: (1) The structural introduction of a dynamic routing mechanism that adaptively adjusts the contributions of multi-scale branches, enhancing the model’s capacity to capture complex spectral patterns. (2) The lightweight grouped multi-scale convolutions combined with an Efficient Channel Attention mechanism significantly reduce the model’s parameter count. (3) To address gradient conflicts among different tasks, uncertainty dynamic weighting is employed to modulate the magnitude of inter-task gradients, further enhancing model stability and predictive accuracy. (4) A soft constraint that the sum of clay, silt, and sand is 100 is introduced into the loss function. The experimental results demonstrate that the effectiveness of the soft constraint is contingent upon the base performance of the model: it effectively promotes inter-task information sharing and significantly enhances model robustness in models with strong base performance; conversely, in models with poorer base performance, it may amplify intrinsic noise, disrupting the training process and degrading performance.

Two datasets, including the large-scale LUCAS (Land Use/Land Cover Area Frame Survey) and the medium-sized ICRAF (International Center for Research in Agroforestry), were used to evaluate the performance of our approach against 11 baseline models. Simultaneously, we validated the effectiveness of a soft-constrained loss term based on prior knowledge and demonstrated the effectiveness of the proposed approach, which incorporates a weighted balance mechanism across tasks.

## 2. Materials and Methods

### 2.1. Soil Dataset Description

Two independent soil datasets were used for modeling: the ICRAF soil dataset and the LUCAS topsoil dataset. Both datasets contain soil physicochemical properties and Vis–NIR spectral data.

The ICRAF (International Center for Research in Agroforestry) soil dataset comprises 4438 soil samples from 58 countries across five continents. The samples were subjected to standardized laboratory processing, including air-drying and passing through a 2 mm sieve. The percentage contents of clay, silt, and sand were determined according to standardized procedures, with values ranging from 0% to 100%. Spectral data were acquired using a FieldSpec FR spectroradiometer, covering a range of 350–2500 nm with a spectral resolution of 1 nm. The LUCAS (Land Use/Cover Area Frame Survey) topsoil database, initiated in 2009, covers 19,036 standardized topsoil samples from 23 EU member states. These samples also underwent rigorous standardization procedures to determine the percentage contents of clay, silt, and sand. Spectral data were obtained in diffuse reflectance mode by using a FieldSpecXDS hyperspectroradiometer. The acquired raw spectra were subsequently interpolated and normalized, resulting in a spectral range of 400–2500 nm with a spectral resolution of 0.5 nm.

Figure 1 illustrates the reflectance and absorbance curves of the raw spectral sequences for the ICRAF and LUCAS datasets used in the analysis.

### 2.2. Preprocessing and Dataset Partitioning

#### 2.2.1. Preprocessing

Data preprocessing was performed to ensure data quality and dimensional consistency. We started by performing quality control and outlier removal on the raw data. Specifically, we removed all samples with missing values. Subsequently, we excluded samples where the sum of the soil texture properties (clay, silt, and sand) was either less than 99% or greater than 101%. In the preprocessing stage, to ensure data accuracy, we applied the Mahalanobis Distance ($D_{M}$) method [40,41] to filter multivariate outliers. This technique effectively accounts for the covariance structure among variables by calculating $D_{M}$ for each sample, sorting them in descending order, and removing the farthest samples. The Mahalanobis Distance threshold (θ) for the ICRAF dataset was set to $\theta_{\mathrm{ICRAF}}=17.7$, resulting in the removal of approximately 2.5% of the samples. For the LUCAS dataset, the threshold was set to $\theta_{\mathrm{LUCAS}}=21.8$, which removed approximately 5% of the samples. This thorough outlier removal process left us with a reliable dataset, with 3501 samples being retained from the ICRAF dataset and 15,762 samples being retained from the LUCAS dataset.

Next, we performed noise reduction and dimension compression on the spectral data. Given the intense noise typically present at the spectral edges, we trimmed 50 nm from each end of the spectra [42]. Subsequently, a Savitzky–Golay (SG) filter [43] with a window size of 19 and a polynomial order of 2 was used for smoothing and noise reduction. To create a compressed spectral representation, all bands were divided into 128 equally sized intervals, and the average spectral value within each interval was calculated.

#### 2.2.2. Dataset Partition

To evaluate model performance, this study employed a 3-fold cross-validation strategy, with the final results reported as the average performance metrics of the 3 folds. To ensure a uniform sample distribution, the K-Means [44] clustering algorithm was applied to the three soil texture properties (clay, silt, and sand), partitioning all samples into 12 clusters, as shown in Figure 2. During dataset construction, samples were allocated to the training and test sets on a cluster-by-cluster basis.

For each cross-validation fold, the dataset was partitioned into three parts: one part served as the test set (33.33% of the total samples), while the remaining two parts served as the modeling set (66.67% of the total samples). Within the modeling set, 20% of the samples were again allocated as the validation set on a cluster-by-cluster basis, with the remaining 80% of the modeling set samples being used as the training set. A visualization of one partition is shown in Figure 3.

### 2.3. Baseline Models

In this study, we employed a diverse set of machine learning and deep learning techniques as benchmarks, selecting 11 representative baseline models. These models, from classic machine learning algorithms to state-of-the-art deep learning architectures, were designed to provide a comprehensive evaluation of the strengths and limitations of different methodological approaches.

For machine learning, we chose Partial Least Squares Regression (PLSR), Support Vector Regression (SVR), Random Forest (RF), Ridge Regression, XGBoost, and LightGBM. These models are widely used for hyperspectral regression tasks due to their efficiency and robustness. All machine learning models were implemented using the scikit-learn 1.6.1 library. For deep learning, we selected several common architectures in the field of hyperspectral data analysis as baselines, including ResNet34, VGG11, Temporal Convolutional Network (TCN) [45], CNN-LSTM [46], and CNN-Transformer [47]. Notably, models like VGG11 and ResNet34 are often used as backbone networks for hyperspectral data processing because of their powerful feature extraction capabilities. These models are widely applied and improved in numerous studies for their excellent performance in spectral data inversion tasks.

### 2.4. MSRA-Net

To address the limitations of existing lightweight models in multi-scale feature fusion and inter-channel information interaction, this study proposes a novel network architecture, Multi-scale Routing Attention Network (MSRA-Net). MSRA-Net is built upon the Multi-scale Routing Attention (MSRA) module as its core building block. This module integrates the Squeeze-and-Excitation (SE) [48] attention mechanism to enhance inter-channel feature interaction, achieving efficient extraction and fusion of multi-scale spectral features through multi-layer stacking. We use maximum pooling to reduce the length of feature sequences, thereby reducing computation overload, as shown in Figure 4.

The model takes a spectral sequence with 128 features as input. The input is first processed by two consecutive 1D convolutional layers (1×3), which expand the number of channels to 32 while extracting preliminary features, which serve to extract features and perform initial noise suppression. Each convolutional layer is followed by a Batch Normalization (BN) layer and a Sigmoid activation layer, which enables the progressive extraction of more abstract features and enhances the model’s nonlinear representational capability.

The model’s end is a Multi-Layer Perceptron (MLP) head with three hidden layers, which maps the features extracted by the backbone network to the final prediction results. The input features are flattened into a vector of length 4096. The number of nodes in each hidden layer of the MLP is reduced by 1/8 from the previous layer, and a Dropout layer (with a random dropout rate of *p* = 0.35) is inserted after each hidden layer to prevent overfitting. The final output layer is a fully connected layer with a length of 3, responsible for the final predictions of clay, silt, and sand percentages.

This section will provide a detailed explanation of the design philosophy and core modules of MSRA-Net and introduce the design of the loss function under uncertainty weighting and prior knowledge soft constraints.

#### 2.4.1. Module Structure

In the design of deep neural networks, a key direction for architectural innovation has been to balance a model’s powerful expressive ability with the need to avoid unnecessary computational redundancy. Selective Kernel Network (SKNet) [49] introduced a soft selection mechanism to achieve the adaptive weighting of features across different convolutional kernels. However, this approach relies on features from branch convolutions to generate weights, which can lead to a potential circular dependency that affects training stability. Subsequently, SkipNet [50] introduced a dynamic skip mechanism into the branch selection process, enabling the network to activate specific layers during inference to reduce computational overhead selectively. However, there is an inconsistency in strategy between the soft gate used during training and the hard gate used during inference. In addition, Dynamic Routing Networks (DRNs) [51] introduced a more refined dynamic routing strategy, allowing the model to automatically determine the optimal information flow path based on the input, significantly enhancing structural adaptability and inference efficiency. Inspired by these works, the proposed framework, MSRA-Net, aims to strike a new balance among the three aspects of multi-scale feature extraction, redundant path compression, and dynamic feature selection.

Shown in Figure 5, the MSRA module is designed to simultaneously enhance the representation capability of multi-scale spectral features and the efficiency of inter-channel information interaction while maintaining a lightweight nature.

To achieve this goal, the MSRA module introduces grouped multi-scale convolutions and an intra-group Efficient Channel Attention (gECA; shown in Figure 6a mechanism, combined with a multi-scale weighting strategy based on a Branch Routing Attention (BRA; illustrated in Figure 6b mechanism, which enables the adaptive selection and fusion of features at different scales.

The two MASR modules divide the channels of the input feature X into four groups of equal size, with 8 and 16 channels in each group. The convolution operation doubles the number of channels to 16 and 32, and the number of channels after splicing in the channel dimension is 64 and 128. It utilizes convolutional kernels of varying lengths (1 × 3, 1 × 5, 1 × 7, 1 × 9) to capture features from receptive fields of different scales and doubles the number of channels, thereby enhancing the model’s multi-scale representation capability. Each convolutional branch is followed by an Efficient Channel Attention (ECA) [39] module.

Channel attention mechanisms are utilized to adaptively recalibrate channel weights, enhancing a model’s feature representation. The Efficient Channel Attention (ECA) module, which efficiently captures local cross-channel interactions via 1D convolution without dimension reduction, is proposed. This design remains lightweight while avoiding information loss. The following formula adaptively determines the size of its convolutional kernel:(1)$$k=\psi \left(C\right)={\left|\frac{\log_{2}\left(C\right)}{\gamma }+\frac{b}{\gamma }\right|}_{odd}$$
where *C* is the number of channels of the input feature map; γ and *b* are two learnable hyperparameters, typically set to 2 and 1, respectively. The convolution kernel size of each ECA convolution in the 2-layer MSRA module is 1×3.

The BRA module receives the input feature X. It first extracts global contextual information through a compression–expansion operation implemented by two linear layers and subsequently generates a set of routing weights. The compression ratio r is 8; that is, the number of nodes in the middle layer is 1/8 of the input layer, a structure commonly employed in multi-branch or Dynamic Routing Networks. These weights adaptively adjust the relative importance of each branch and enhance their contributions to the final prediction.

To prevent the branch weights from collapsing too early during training, a higher-temperature Softmax is employed in the initial stage to ensure that all branches are activated. The pre-relaxation mechanism facilitates free exploration among branches. As training progresses into the middle and later stages, a linear annealing strategy is adopted to gradually decrease the temperature [52], resulting in a sharper weight distribution and thus enabling more deterministic branch selection. The normalization is defined as follows:(2)$$\alpha_{i}=\frac{\exp(z_{i}/T)}{\sum_{j=1}^{n}\exp\left(z_{j}/T\right)}$$
where *T* is the temperature coefficient, which decreases from 5.0 to 0.5 during training (linear annealing strategy).

After the grouped convolutions, the feature sequences from all groups are concatenated along the channel dimension, effectively doubling the number of channels compared with the input. The concatenated features are then processed by a 1×3 convolution followed by BN and a ReLU activation layer, enhancing local spectral feature representation while maintaining training stability. To facilitate residual learning, a 1×1 convolution is applied to the input to match the channel dimension, and a residual connection is added. Subsequently, the features are fed into an Squeeze-and-Excitation (SE) block (as shown in Figure 7) to recalibrate channel-wise importance, emphasizing informative spectral channels.

Finally, a max-pooling operation is applied to reduce the spatial dimension and aggregate dominant features. In this network, two MSRA modules further expand the number of channels to 64 and 128. After max pooling, the lengths of the feature sequences are reduced to 64 and 32, respectively.

#### 2.4.2. Loss Function

In the multi-task learning (MTL) area, inconsistencies in the magnitudes of different task losses often lead to conflicting gradient directions or uneven gradient scales, thereby impacting model convergence and generalization performance. To address this challenge, we proposed a dynamic loss weighting method based on homoscedastic uncertainty [53], in which each task is referred to as a regression problem with a distinct observation noise variance and dynamically adjusts its weight in the total loss based on this variance. Tasks with larger noise variance are given lower weights, which achieves a balance in the optimization of each task and automatically adapts to their relative importance.

Assuming the observation noise variance for the $t^{th}$ task is $\sigma_{t}^{2}$. The dynamically weighted total loss based on homoscedastic uncertainty can be written as(3)$$L_{\mathrm{uncertainty}}\left(W,\sigma_{1},\ldots ,\sigma_{T}\right)=\sum_{t=1}^{T}\frac{1}{2\sigma_{t}^{2}}L_{t}\left(W\right)+\log\left(\sigma_{t}\right)$$
where $L_{t}\left(W\right)$ is the Mean Squared Error (MSE) loss for the $t^{th}$ task.

When a task exhibits high inherent error or noise, the learned observation noise variance $\sigma_{t}^{2}$ automatically increases, thereby preventing that task from dominating the training process. In this context, $\log(\sigma_{t})$ serves as a regularization term constraining $\sigma_{t}^{2}$ from growing unboundedly, which would degrade the loss to zero weight. Prior to training, $\sigma_{t}$ is typically initialized to 1.0.

Simultaneously, to leverage prior knowledge of soil texture—that is, the sum of clay, silt, and sand should equal 100%—we introduce a soft constraint penalty term:(4)$$L_{prior}=\lambda \;{∥\mathrm{clay}+\mathrm{silt}+\mathrm{sand}-100∥}_{1}$$

λ is a weighting coefficient that controls the influence of the prior knowledge constraint on the overall loss. This term is introduced in the form of an L1 penalty.

Finally, the overall loss function is defined as(5)$$L_{total}=L_{uncertainty}+L_{prior}$$

### 2.5. Experimental Design and Evaluation Metrics

#### 2.5.1. Experimental Design

In this study, the MSRA-Net architecture was adopted as the base model, developing two versions: a single-task version (designated as -ST) and a multi-task version (designated as -MT). The two variants differ in the number of nodes in the final fully connected layer and in their loss functions. Specifically, the single-task model’s output layer contains a single node, while the multi-task model includes multiple nodes corresponding to the number of prediction tasks. Other baseline models (e.g.,VGG11 and ResNet34) were configured with the same settings as MSRA for task heads and training procedures across both variants. In this research study, since prediction targets are the percentages of clay, silt, and sand, the multi-task output layer consists of three nodes. Furthermore, the multi-task version is optimized using uncertainty weighting and a soft constraint loss function based on prior knowledge, whereas these components are omitted in the single-task version. To ensure a fair comparison, all baseline models were implemented following the same design principles.

Three-fold cross-validation is employed for evaluating model performance. In each fold, models are trained on the training set, and the best-performing model is saved based on the average RMSE across all tasks on the validation set. After training, the models are evaluated on the test set, and we report the mean performance over the three folds for both individual tasks and the overall average performance of the three tasks.

For the traditional machine learning models, a slightly different procedure was followed. We used a grid search to identify the best hyperparameters on the validation set. Afterward, the models were retrained on the combined training and validation sets (the modeling set), and their average results on the test set were reported.

The MSRA-Net models were trained using the Adam optimizer with a learning rate of $1.85\times 10^{-3}$. We used a cosine annealing learning rate scheduler (CosineAnnealingLR) with T_max set to 500 and eta_min to $5\times 10^{-4}$. To ensure the optimal performance of the proposed model, key hyperparameters were automatically determined using the Optuna hyperparameter optimization framework [54]. Optuna employs Bayesian optimization coupled with the Tree-structured Parzen Estimator (TPE) algorithm, allowing for efficient exploration within a predefined search space to converge on the best-performing parameter set. The boundaries of this search space were defined with comprehensive consideration of the model’s training efficiency and computational cost. The objective function for this optimization process was set as the minimization of the Mean Squared Error (MSE) on the validation set. The batch size was set to 96 for the ICRAF dataset and 128 for the LUCAS dataset. All models were trained for 500 epochs. The learning rate for the noise standard deviation parameter $\sigma_{t}$ in the loss function was set to $5\times 10^{-4}$, and the regularization coefficient λ for the $L_{prior}$ loss term was also set to $5\times 10^{-4}$. The open source code is published on https://github.com/surpriseac/MRSE-Net accessed on 20 October 2025.

We fixed the global random number seeds for Python, NumPy, and PyTorch and disabled CuDNN’s automatic selection of optimization algorithms by setting torch.backends.cudnn.deterministic to True. This ensures complete determinism in the training process in the same hardware and software environment. All experiments were conducted on an Ubuntu 22.04 system equipped with an Intel Xeon E5-2696 v3 processor and an NVIDIA Titan Xp GPU. The models were developed using Python 3.10.18 and the PyTorch 2.4.1 framework.

#### 2.5.2. Evaluation Metrics

This study employs four standard metrics to evaluate the model’s overall performance. The Root Mean Square Error (RMSE) quantifies the average deviation between predicted and actual values. The coefficient of determination ($R^{2}$) represents the proportion of the variance in the dependent variable that can be explained by the independent variable(s). Its value ranges from 0 to 1, with values closer to 1 indicating a better model fit. The Relative Prediction Deviation (RPD) is the ratio of the standard deviation (SD) to the RMSE, commonly used to assess the robustness of model predictions. The Mean Absolute Error (MAE) is the average of the absolute differences between predicted and actual values. Compared with the RMSE, the MAE is less sensitive to outliers. These metrics are defined by Equations (Equation 6)–(Equation 9) respectively.(6)$$\mathrm{RMSE}=\sqrt{\frac{1}{n}\sum_{i=1}^{n}{\left(y_{i}-{\hat{y}}_{i}\right)}^{2}}$$(7)$$R^{2}=1-\frac{\sum_{i=1}^{n}{\left(y_{i}-{\hat{y}}_{i}\right)}^{2}}{\sum_{i=1}^{n}{\left(y_{i}-\bar{y}\right)}^{2}}$$(8)$$\mathrm{RPD}=\frac{\mathrm{SD}}{\mathrm{RMSE}}=\frac{\sqrt{\frac{1}{n-1}\sum_{i=1}^{n}{\left(y_{i}-\bar{y}\right)}^{2}}}{\mathrm{RMSE}}$$(9)$$\mathrm{MAE}=\frac{1}{n}\sum_{i=1}^{n}\left|y_{i}-{\hat{y}}_{i}\right|$$
where *n* is the total number of samples; $y_{i}$ and $\hat{y_{i}}$ are the actual and predicted values for the *i*-th sample, respectively; $\bar{y_{i}}$ is the mean of all actual values; and SD is the standard deviation of the actual values.

## 3. Results

### 3.1. Model Performance

Table 1 illustrates the average performance of the MSRA-Net model’s multi-task version (MSRA-MT) and single-task version (MSRA-ST) when predicting soil texture properties (clay, silt, and sand) on the ICRAF-ISRIC and LUCAS-ESDAC datasets. The reported values, including RMSE, R^2^, RPD, and MAE, are the average results from 10 independent runs with different random seeds, including the means and standard deviations of the four evaluation indicators of the three tasks.

The multi-task learning paradigm of the MSRA model (MSRA-MT) outperforms the single-task version (MSRA-ST) in both performance and stability, demonstrating the ability to leverage the intrinsic correlations among tasks effectively. Experimental results based on the ICRAF and LUCAS datasets are in accordance with the conclusion. On the ICRAF dataset, the Mean Squared Error (RMSE) of the MSRA-MT for the Clay, Silt, and Sand prediction tasks was reduced by 2.8%, 3.9%, and 7.8%, respectively. Similarly, on the LUCAS dataset, the RMSE was reduced by 2%, 7.2%, and 3.2%, respectively. On average, the RMSE of the MSRA-MT was reduced by 5.22% on the ICRAF dataset and 4.44% on the LUCAS dataset. Concurrently, the standard deviation of the average metrics for the MSRA-MT version was also significantly lower than that of MSRA-ST, indicating that the proposed model’s prediction results are more stable. To examine the performance difference between MSRA-MT and MSRA-ST, we performed a paired *t*-test on the results of 10 repeated experiments in two groups. For the ICRAF and LUCAS datasets, the t-statistic values corresponding to the mean difference in RMSE for the three tasks were −9.223 (*p*-value < 0.001) and −27.537 (*p*-value < 0.001), respectively. The t-values corresponding to the mean difference in $R^{2}$ were 11.005 (*p*-value < 0.01) and 19.846 (*p*-value < 0.001), respectively.

Figure 8 illustrates the results of a repeated experiment on the MSRA-MT model with an uncertainty-weighted and prior knowledge-based soft-constrained loss function; the curves represent the average RMSE values for the validation sets of the three tasks. Figure 9 presents a scatter plot of predicted versus observed values for the median performance from a repeated experiment. This figure superimposes the results from the 3-fold cross-validation.

The training curves indicate that the MSRA-MT model’s performance improved rapidly after approximately 100 epochs. Subsequently, the performance curves stabilized around 350 epochs, eventually converging to a steady level. Furthermore, the 95% confidence interval was wider during the early training stages, reflecting the volatility in the model’s performance. However, during the convergence phase, the interval narrowed significantly, indicating that the model’s performance had become robust and stable.

Significant improvement can be attributed to the model’s intrinsic mechanism. By sharing parameters and undergoing co-training, the MSRA-MT model effectively captures the underlying commonalities of different soil texture properties in their spectral response signals. The approach facilitates positive knowledge transfer among tasks, enabling the model to generate a synergistic gain effect. Consequently, this not only enhances the model’s prediction accuracy and generalization capability but also ensures the robustness and reliability of the results.

### 3.2. Comparative Experiment

#### 3.2.1. Comparison of Multi-Task and Single-Task Learning Models

To comprehensively validate the effectiveness of the two loss function optimization methods we proposed, we designed a series of comparative experiments. We integrated these methods with our MSRA-MT model and five other multi-task baseline models. Each model was then evaluated under three distinct loss function configurations:(1)Static Weighting: Average of the Mean Squared Error (MSE) losses of the three prediction tasks.(2)Uncertainty Dynamic Weighting: Its definition is shown in Equation (Equation 3).(3)Static Weighting with Soft Constraints: Based on static weighting, the soft constraint loss term Lprior is introduced, which is defined as shown in Equation (Equation 4).

All experiments reported the mean values of the Root Mean Squared Error (RMSE) and the coefficient of determination ($R^{2}$) to evaluate the models’ performance comprehensively. The detailed experimental results are presented in Table 2.

Experimental results on both datasets reveal a significant commonality: almost all models that used an uncertainty-based dynamic weighting form of loss function outperformed those with traditional static average weighting. This performance difference was particularly evident in the MSRA-MT model. However, not all models benefit from the $L_{prior}$ soft constraint loss term. Notably, models with inherently poorer performance (such as ResNet34-MT and LSTM-MT) exhibited a further decline in performance after incorporating the soft constraint loss term. To meticulously analyze the mechanism of the soft constraint loss term ($L_{\mathrm{prior}}$) on various models and individual tasks, we plotted the change in the RMSE for repetitive experiments across the three tasks, Clay, Silt, and Sand, for six models (five multi-tasks baseline models and the MSRA-MT model) after the introduction of $L_{\mathrm{prior}}$. The RMSE variation (ΔRMSE) is defined such that ΔRMSE>0 indicates an increase in the model’s prediction loss after the addition of the soft constraint loss term. Figure 10 illustrates the ΔRMSE of the six models across the three tasks on two different datasets.

The analysis of the ΔRMSE data in Figure 10 reveals a critical phenomenon: the impact of the prior knowledge soft constraint loss term ($L_{\mathrm{prior}}$) on the prediction performance of individual tasks exhibits a consistent and fixed hierarchy of strength. Across nearly all evaluated models, regardless of whether $L_{\mathrm{prior}}$ results in performance gain (ΔRMSE<0) or degradation (ΔRMSE>0), the magnitude of the effect, quantified by the absolute ΔRMSE (|ΔRMSE|), maintains an identical order: **Sand > Silt > Clay**. The Sand task is consistently the most strongly affected, while the Clay task shows the minimum change.

This consistent hierarchy strongly suggests that the intrinsic uncertainty and noise level of the individual tasks govern the distribution of the $L_{\mathrm{prior}}$ correction. Specifically, the prediction of sand is inherently the most uncertain (or noise-prone) due to its weaker spectral characteristics. Consequently, to minimize the overall closure error of 100%, the multi-task model preferentially allocates the largest corrective or optimization gradient updates to the most volatile task (Sand), where the correction yields the maximum reduction in the soft constraint loss. While this strategy aims for the most effective overall error minimization, it simultaneously causes the optimization process to be dominated by the noisy task, leading to potential optimization conflict among the tasks. This finding highlights the necessity of employing an inter-task optimization conflict mitigation strategy to ensure that the constrained optimization does not disproportionately sacrifice the performance of one task for the benefit of the constraint.

This suggests that a soft constraint is not universally beneficial for all models. We hypothesize that for models with weak initial performance, the structural residuals of their predictions (i.e., abs(clay + silt + sand − 100)) inherently contain more noise and systematic bias. In this case, introducing the L prior soft constraint not only fails to correct the model effectively but may also amplify this inherent noise as an additional loss term, thereby interfering with the optimization process. Therefore, when using prior knowledge-based soft constraints to assist training, a model’s initial performance and intrinsic robustness are crucial.

A core challenge in multi-task learning (MTL) is task conflicts. While shared parameters can theoretically promote knowledge sharing, in practice, when task objective functions are contradictory, the optimization process may lead to negative transfer, thereby degrading overall model performance. To validate this hypothesis and provide a benchmark for subsequent conflict mitigation strategies, we conducted experiments on the single-task versions of MSRA-ST and five baseline models (VGG11-ST, ResNet34-ST, TCN-ST, LSTM-ST, and Transformer-ST). Each model was trained independently for the three tasks of Clay, Silt, and Sand. We also experimented with six commonly used traditional machine learning models, with their average performance reported as shown in Table 3.

Among the 12 models evaluated, deep learning models consistently outperformed traditional machine learning models on two datasets. Among the baseline models, Transformer-ST and TCN-ST performed optimally. Both models effectively addressed the challenge faced by traditional models in capturing long-range dependencies within complex spectral data. In contrast, traditional machine learning methods showed mediocre performance on the soil texture prediction task, with an RDP value of less than 2.0.

It is noteworthy that the performance of these conventional machine learning models (e.g., [mention the traditional models you used, such as SVR, PLSR]) is generally poor (RDP<2.0). This is primarily due to the inherent limitations of these models when dealing with high-dimensional, highly collinear, and non-linear hyperspectral data. They typically rely on linear or shallow feature engineering (e.g., principal component analysis or feature selection) to process thousands of bands, making it difficult for them to automatically learn complex, abstract feature representations from the high-dimensional non-linear spectral data. Consequently, they struggle to capture the complex non-linear mapping relationship between soil constituents and spectral reflectance. In contrast, deep learning models can achieve progressive feature abstraction and representation learning through their multi-layer non-linear structures, which is key to their superior performance in hyperspectral prediction tasks.

A comparative analysis was conducted between the deep learning models, including MSRA-ST, and the multi-task models using a static weighting scheme from Table 2. The results indicate that the multi-task models did not exhibit the expected performance advantages on the three tasks without any conflict mitigation strategies. Their average performance was even slightly inferior to that of their corresponding independent single-task models. This outcome confirms the presence of task conflict within the models, which negatively impacted their overall performance.

#### 3.2.2. Uncertainty Dynamic Weighting or Static Weighting?

Uncertainty dynamic weighting is adopted as the optimization strategy for the loss function. The core idea of this strategy is to use the predictive uncertainty of each task as the weight for its corresponding loss term, allowing the model to adaptively adjust its focus on different tasks during training, thereby mitigating inter-task conflicts. To validate the effectiveness of this approach, we compared the gradient cosine similarity of the three tasks during the training process of the MSRA-MT model, under both static and dynamic weighting schemes. We computed the gradients of each task with respect to the last MSRA module and SE module. These high-dimensional gradient tensors were then flattened into one-dimensional vectors. Subsequently, the cosine similarity among these three gradient vectors was calculated. As shown in Figure 11, the curves represent the mean values from a 3-fold cross-validation, and the translucent shaded regions indicate the 95% confidence intervals.

The cosine similarity reflects the similarity of two tasks’ gradient directions, with a range from −1.0 to 1.0:Cosine = 1.0 indicates that the gradient directions are identical, showing high synergy between tasks.Cosine = 0.0 indicates that the gradients are orthogonal, meaning that the updates do not affect each other.Cosine = −1.0 indicates that the gradient directions are opposite and that the updates cancel each other out.

Based on the analysis of the MSRT-MT model on the ICRAF and LUCAS datasets, a common issue in multi-task learning was identified: In the baseline model without inter-task optimization, a negative transfer existed between the Clay and Sand tasks and the Silt task, with their gradients exhibiting a competitive tug-of-war. However, under two weighting schemes, the Clay and Sand tasks themselves showed no significant conflict. This suggests that without an effective coordination mechanism, multi-task learning can face considerable conflicts among some of its tasks.

Furthermore, upon introducing the uncertainty-based weighting method, the cosine similarity of the gradients for all task pairs consistently converged to a positive value during training. Although the similarity value was not high, this is sufficient to prove the strategy’s effectiveness. It not only successfully transformed the negative transfer into positive cooperative optimization but also ultimately led to a significant performance improvement for MSRT-MT on all three tasks. The details of these improvements are presented in Table 4 and Table A1.

### 3.3. Ablation Experiments

To evaluate the performance of three key components of the MSRA model (ECA, SE, and BRA), we sequentially removed each module from the complete model (referred to as the “Standard” model). This process was followed by ten repeated experiments to observe the resulting performance degradation. The experiments were conducted on both the ICRAF and LUCAS datasets, and the results are presented in Table 5 and Table A2.

The experimental results indicate that removing the BRA module resulted in the most significant performance degradation for both datasets, as evidenced by the highest mean RMSE and MAE values. The model with the BRA module removed performed worse than the models with the ECA or SE module removed. Furthermore, the standard deviation of its mean RMSE in repeated experiments was 47% and 34% higher than that of the complete model, highlighting the critical role of the BRA module in the model’s robust performance. While the ECA and SE modules also played a role, their removal led to a less severe decline in performance, suggesting that they provide additional rather than fundamental enhancements to the model’s feature extraction capabilities.

## 4. Discussion

This paper proposed a lightweight multi-scale convolutional model, MSRA-Net, and its two variants, single-task (ST) and multi-task (MT), for individually and simultaneously predicting multiple soil texture properties (clay, silt, and sand). We employ uncertainty dynamic weighting to balance the optimization of different tasks and utilize prior knowledge as a soft constraint to guide the model in learning inter-task relationships.

### 4.1. Multi-Task Learning or Single-Task Learning?

Repeated three-fold cross-validation experiments on two public different scales datasets, ICRAF and LUCAS, demonstrate that MSRA-MT outperforms its single-task counterparts, other baseline models, and their multi-task variants. Specifically, on the ICRAF dataset, the R^2^ for clay, silt, and sand prediction reaches 0.875, 0.780, and 0.831, respectively. On the LUCAS dataset, the $R^{2}$ values are 0.863, 0.735, and 0.787. Compared with the single-task variant, the average multi-task RMSE of MSRA-MT is reduced by 5.22% and 4.44% on the ICRAF and LUCAS datasets, respectively. Compared with the traditional single-task learning (STL) architecture, the multi-task learning (MTL) framework adopted by MSRA-MT offers significant methodological and practical advantages, particularly for large-scale and heterogeneous remote sensing data processing. The core benefit of MTL lies in enforced feature sharing and enhanced generalization: it requires the model to share the underlying feature representation across all related tasks (sand, silt, and clay contents), thereby compelling the model to learn a more generic and robust soil spectral pattern, rather than local noise or biases specific to a single grain size component. This ultimately and effectively addresses the critical bottlenecks faced by conventional STL models in remote sensing spectral data, namely, the sparsity of labeled samples and the lack of generalization capability (i.e., overfitting) in heterogeneous environments.

By training five baseline models based on uncertainty dynamic weighting and the prior knowledge soft constraint loss terms, we found that multi-task learning models face significant conflicts across specific tasks without an effective coordination mechanism. Uncertainty dynamic weighting effectively mitigates the issue and significantly enhances the stability of performance in repeated experiments. Furthermore, the prior knowledge soft constraint has a dual effect on model performance: it is beneficial for models with excellent initial performance. However, it can be detrimental to models with weaker initial performance. Ablation experiments show that the Branch Routing Attention (BRA) module in MSRA-MT contributes most significantly, validating its effectiveness in integrating multi-scale information and enhancing feature representation. The intra-group Efficient Channel Attention (gECA) and Squeeze-and-Excitation (SE) modules also provide an inevitable performance gain to the proposed model.

### 4.2. Prospects and Limitations

The proposed model, MSRA-Net, demonstrates superior performance on two different scale datasets, providing an efficient and reliable solution for the accurate prediction of soil texture properties. The lightweight design and multi-task learning architecture can effectively reduce computational demands, thereby meeting the requirements for low latency and low power consumption on on-board edge computing devices such as UAVs or handheld spectrometers. This ensures the model’s capability for the fast, real-time acquisition and processing of soil texture information.

The MTL architecture naturally integrates the three tasks into a single optimization space, resulting in predictions with greater physical plausibility than independent STL models. This robustness is crucial to processing multispectral and hyperspectral satellite imagery (such as Sentinel-2 and Landsat). Once trained, the model can be directly applied to remote sensing data from vast regions to extract soil texture information efficiently. This is crucial to large-scale environmental issues such as regional soil weathering, land degradation, and desertification monitoring. It can quickly and accurately identify soil texture trends, providing data support for macro-level soil and water conservation strategies.

The uncertainty dynamic weighting (UDW) mechanism introduced in MSRA-MT is key to mitigating this conflict. This mechanism dynamically adjusts the penalty weights of each task by assessing its inherent uncertainty, essentially balancing the magnitude of conflicting gradients. This task-agnostic mechanism offers a viable solution for future applications to other soil nutrient prediction tasks. It ultimately enables the construction of a unified model for high-precision prediction of core tasks such as pH, organic carbon, nitrogen, phosphorus, potassium, salinity, and heavy metals.

However, since this method was developed based on large-scale spectral libraries collected under laboratory conditions, its future deployment in field environments will confront complex spectral interferences. Furthermore, the model may encounter Domain Shift when deployed across different times and geographical regions. Therefore, future work will focus on integrating the model with advanced data preprocessing techniques or domain adaptation strategies to ensure that MSRA-Net can maintain or even surpass its current performance on high-variability, large-scale UAV-collected data. On the other hand, soil texture is one of the main covariates influencing soil spectral properties. It indirectly affects the spectral signal of soil nutrients by influencing soil moisture, porosity, specific surface area, and aggregate structure. Inaccurate texture information is a significant factor contributing to the poor cross-regional transfer performance of soil nutrient models. By accurately and interpretably extracting soil texture information through MSRA-MT, we can use the predicted texture parameters or the model’s internal texture representation as auxiliary inputs or constraints for domain adaptation. This can alleviate the paradox of poor soil nutrient model transfer performance due to soil texture differences, significantly improving the robustness and accuracy of cross-regional soil nutrient remote sensing estimation.

In conclusion, this research study provides a valuable reference for identifying potentially degraded areas and formulating differentiated governance measures, laying a solid foundation for the promotion and application of this method at field and remote sensing scales.

## Figures and Tables

**Figure 1 sensors-25-06519-f001:**
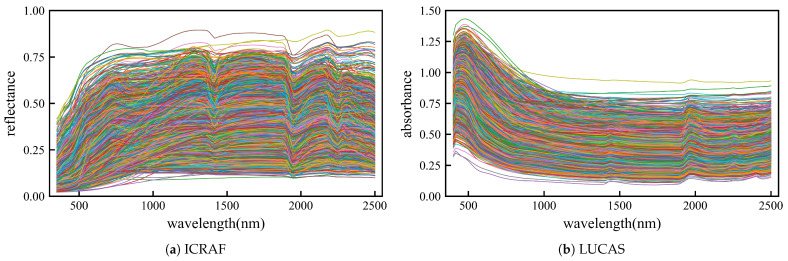
Original spectral reflectance curve. (**a**) curve of ICRAF-ISRIC dataset (3501 samples). (**b**) curve of LUCAS-ESDAC dataset (15,762 samples).

**Figure 2 sensors-25-06519-f002:**
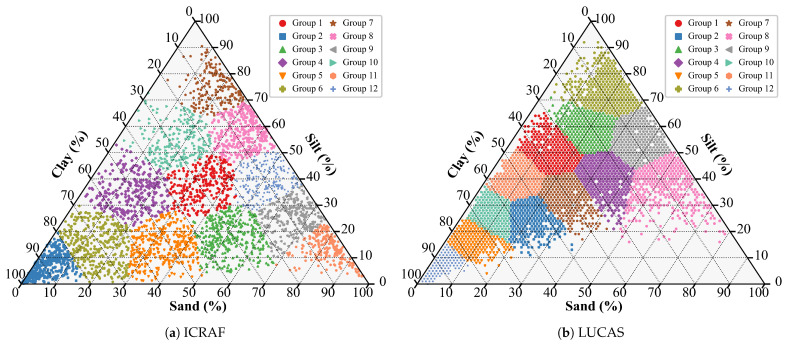
Soil texture triangle with K-Means clustering results (12 clusters).

**Figure 3 sensors-25-06519-f003:**
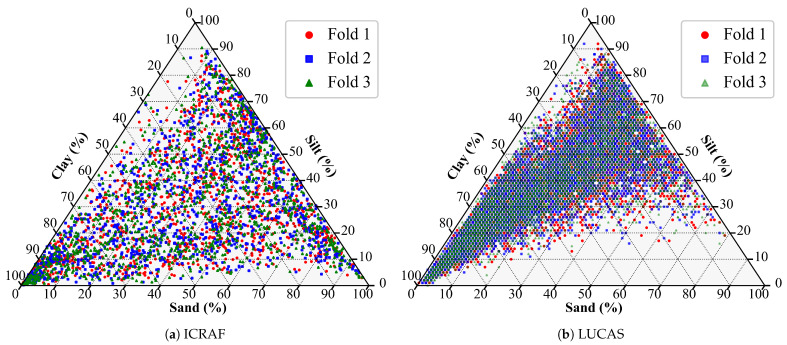
Soil texture triangle of one training/test set partition.

**Figure 4 sensors-25-06519-f004:**
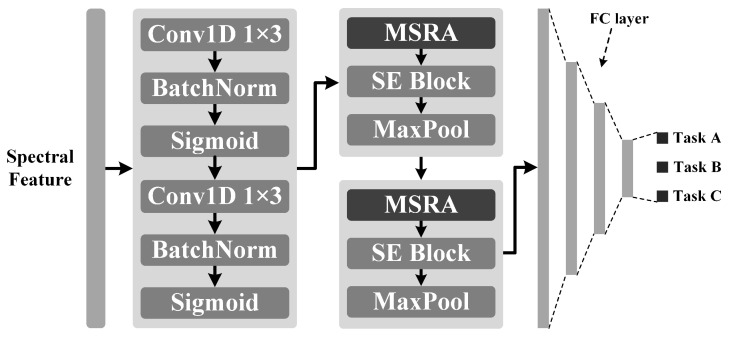
The overall architecture of Multi-Scale Routing Attention Network (MSRA-Net).

**Figure 5 sensors-25-06519-f005:**
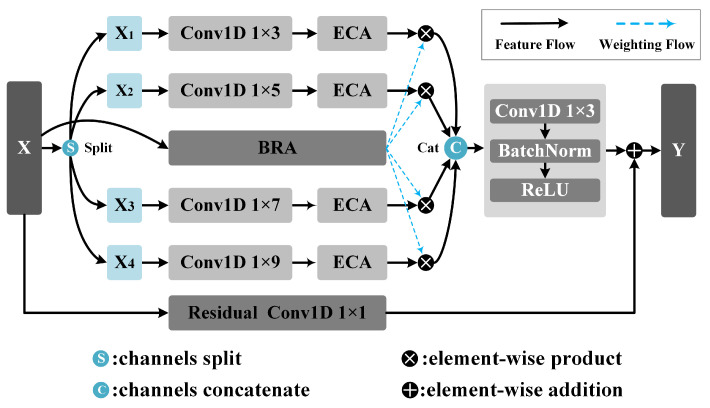
The structure of the Multi-Scale Routing Attention (MSRA) module.

**Figure 6 sensors-25-06519-f006:**
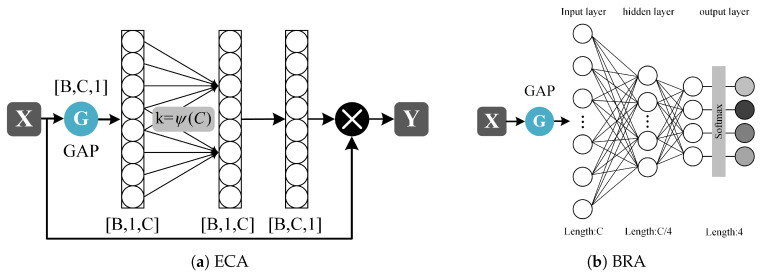
The structure of ECA and BRA modules. (**a**) the Efficient Channel Attention (ECA) modules. (**b**) the Branch Routing Attention (BRA) modules.

**Figure 7 sensors-25-06519-f007:**
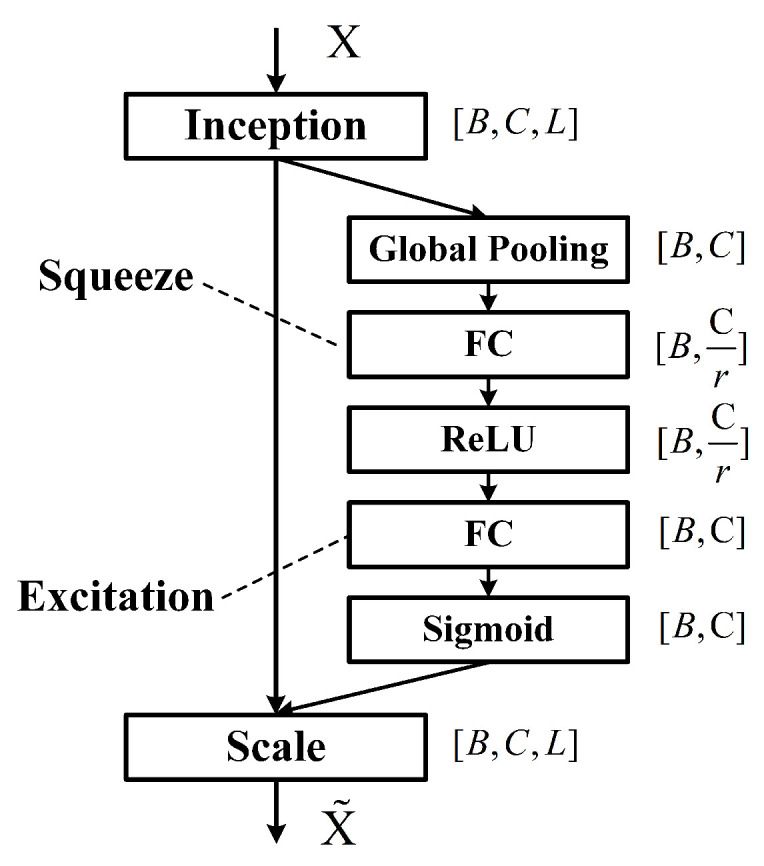
Squeeze-and-Excitation (SE) block structure for channel-wise feature recalibration. The reduction ratio is set to r=8.

**Figure 8 sensors-25-06519-f008:**
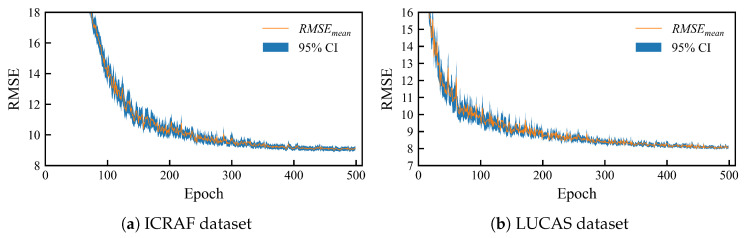
Training convergence curves of the MSRA-MT model. (**a**) from the ICRAF-ISRIC dataset and (**b**) from the LUCAS-ESDAC dataset. The solid lines represent the average RMSEs of the three tasks, and the semi-transparent intervals are the 95% confidence intervals.

**Figure 9 sensors-25-06519-f009:**
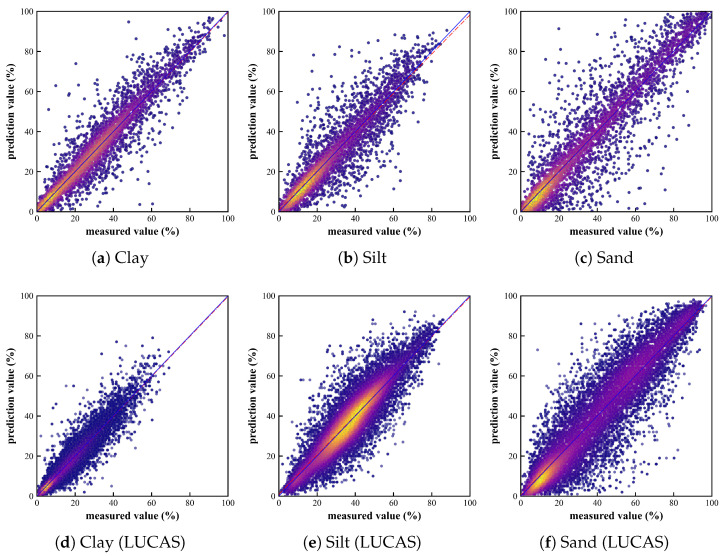
Scatter plots of predicted vs. observed values for three tasks by MSRA-MT-Net. (**a**–**c**) ICRAF-ISRIC dataset, showing results for: (**a**) clay; (**b**) silt; and (**c**) sand. (**d**–**f**) LUCAS-ESDAC dataset, showing results for: (**d**) clay; (**e**) silt; and (**f**) sand. The plot superimposes the prediction results from all three folds of the cross-validation. The X-axis and Y-axis correspond to the measured value and the predicted value, respectively, ranging from 0% to 100%. The blue lines represent the 1:1 lines, and the red lines represent the regression lines.

**Figure 10 sensors-25-06519-f010:**
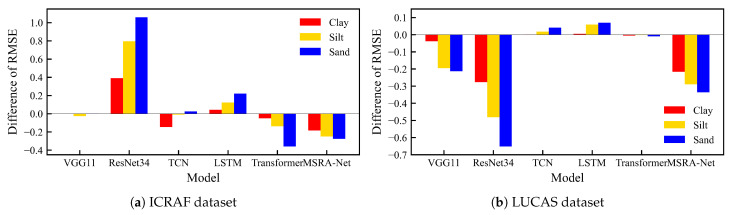
Impact of the soft constraint loss term ($L_{\mathrm{prior}}$) on the RMSE for the three prediction tasks across different models. On the vertical axis, a positive value (ΔRMSE>0) indicates an increase in RMSE (performance degradation) after adding the soft constraint term.

**Figure 11 sensors-25-06519-f011:**
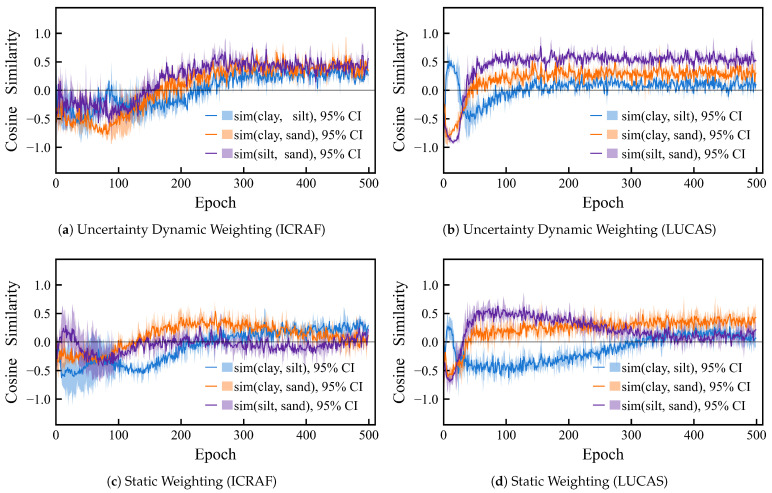
Comparison of inter-task gradient cosine similarity training curves under different weighting strategies, uncertainty dynamic weighting vs. static weighting. (**a**) Uncertainty dynamic weighting on the ICRAF-ISRIC dataset; (**b**) Uncertainty dynamic weighting on the LUCAS-ESDAC dataset; (**c**) Static weighting on the ICRAF-ISRIC dataset; (**d**) Static weighting on the LUCAS-ESDAC dataset. The solid lines represent the means, and the semi-transparent intervals are the 95% confidence intervals.

**Table 1 sensors-25-06519-t001:** Performance of the MSRA-MT-Net multi-task regression model in soil texture prediction.

Model	Task	ICRAF				LUCAS			
		RMSE	$R^{2}$	RPD	MAE	RMSE	$R^{2}$	RPD	MAE
MSRA-MT	Clay	7.581	0.883	2.928	5.087	4.630	0.872	2.792	3.225
	Silt	8.897	0.808	2.282	5.932	8.769	0.772	2.094	6.512
	Sand	11.091	0.854	2.624	7.335	11.169	0.818	2.343	8.329
	Mean ^1^	9.190	0.848	2.611	6.118	8.189	0.820	2.410	6.022
	SD ^2^	0.0682	0.0018	0.0189	0.0418	0.0434	0.0018	0.0124	0.0407
MSRA-ST	Clay	7.803	0.876	2.846	5.215	4.725	0.865	2.726	3.265
	Silt	9.252	0.792	2.196	6.180	9.444	0.735	1.943	7.004
	Sand	12.035	0.828	2.422	8.049	11.536	0.806	2.273	8.522
	Mean	9.697	0.832	2.488	6.481	8.568	0.802	2.314	6.264
	SD	0.0790	0.0025	0.0216	0.0461	0.0020	0.0021	0.0148	0.0444

^1^ The mean values are the averages of the three tasks (Clay, Silt, and Sand). ^2^ The SD values are the standard deviations of the mean values obtained from multiple experimental runs.

**Table 2 sensors-25-06519-t002:** Multi-task model performance with different loss functions and weighting methods.

Model	Loss Item	ICRAF		LUCAS	
		RMSE	$R^{2}$	RMSE	$R^{2}$
VGG11-MT	StaticWeighting ^1^	10.509	0.804	9.525	0.757
	Uncertainty ^2^	10.379	0.808	9.532	0.756
	Static&Prior ^3^	10.481	0.805	9.375	0.764
ResNet34-MT	StaticWeighting	11.804	0.753	10.774	0.689
	Uncertainty	11.582	0.762	11.230	0.660
	Stati&Prior	12.552	0.717	10.304	0.716
TCN-MT	StaticWeighting	10.520	0.803	9.427	0.762
	Uncertainty	10.487	0.804	9.307	0.768
	Static&Prior	10.475	0.804	9.446	0.761
LSTM-MT	StaticWeighting	14.068	0.650	10.386	0.710
	Uncertainty	14.515	0.629	10.136	0.719
	Static&Prior	14.197	0.644	10.431	0.707
Transformer-MT	StaticWeighting	10.144	0.816	8.772	0.788
	Uncertainty	9.961	0.823	8.706	0.796
	Static&Prior	10.065	0.819	8.774	0.793
MSRA-MT	StaticWeighting	9.777	0.828	8.763	0.795
	Uncertainty	9.374	0.842	8.353	0.814
	Static&Prior	9.518	0.837	8.482	0.807

^1^ Use of only static weighting: $\mathrm{Loss}=\frac{1}{n}\sum_{n}^{t=1}L_{t}\left(W\right)$. ^2^ Use of only uncertainty dynamic weighting method. ^3^ Addition of prior knowledge soft constraints on the basis of static weighting: $\mathrm{Loss}=\frac{1}{n}\sum_{n}^{t=1}L_{t}\left(W\right)+L_{prior}$.

**Table 3 sensors-25-06519-t003:** Single-task model performance.

Model	ICRAF				LUCAS			
	RMSE	$R^{2}$	RPD	MAE	RMSE	$R^{2}$	RPD	MAE
PLSR	16.106	0.543	1.495	12.721	11.359	0.654	1.744	8.802
Ridge	16.041	0.547	1.502	12.654	11.907	0.620	1.669	9.333
SVR	13.478	0.673	1.810	9.733	11.985	0.620	1.671	9.576
RF	16.723	0.508	1.445	12.940	14.087	0.464	1.370	11.023
XGBoost	16.590	0.515	1.457	12.520	13.712	0.492	1.410	10.599
LightGBM	16.376	0.528	1.480	12.345	13.701	0.491	1.411	10.557
VGG11-ST	10.455	0.804	4.424	4.955	9.717	0.748	2.058	7.232
ResNet34-ST	11.304	0.773	2.136	7.555	10.033	0.734	1.991	7.355
TCN-ST	10.343	0.809	2.327	6.977	9.324	0.768	2.134	6.887
LSTM-ST	13.877	0.661	1.741	9.973	10.564	0.706	1.913	7.961
Transformer-ST	10.194	0.814	2.377	6.909	8.879	0.789	2.212	6.591
MSRA-ST	9.697	0.832	2.488	6.481	8.568	0.802	2.314	6.264

**Table 4 sensors-25-06519-t004:** Mean performance of MSRA-MT model under two task-weighting strategies.

Weighting Method	ICRAF				LUCAS			
	RMSE	R^2^	RPD	MAE	RMSE	R^2^	RPD	MAE
Static	9.777	0.828	2.463	6.333	8.763	0.795	2.271	6.382
Uncertainty	9.374	0.842	2.565	6.243	8.353	0.814	2.366	2.366

**Table 5 sensors-25-06519-t005:** Mean performance of the model after removing different modules.

Dropped Module	ICRAF				LUCAS			
	RMSE	R^2^	RPD	MAE	RMSE	R^2^	RPD	MAE
Standard	9.1898	0.8483	2.6113	6.1181	8.1890	0.8205	2.4096	6.0220
Drop ECA	9.4349	0.8407	2.5469	6.3065	8.3547	0.8119	2.3592	6.0866
Drop SE	9.4530	0.8400	2.5417	6.2951	8.4295	0.8091	2.3440	6.2260
Drop BRA	9.7164	0.8317	2.4718	6.4341	8.6419	0.7995	2.2831	6.3140

## Data Availability

The data used for this study can be found at The Open Soil Spectral Library [55,56] (OSSL), a compilation of several heterogeneous and independent datasets into a common standardized source. It is available as a digital asset and web-service. It contains standardized data from the ICRAF and LUCAS datasets.

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
