# Peer review of "MSRA-Net: A Multi-Task Learning Model for Soil Texture Prediction with Dynamic Weighting and Prior Knowledge Soft Constraints"

_sensors, 2025, doi:10.3390/s25216519_

Round 1

Reviewer 1 Report

Comments and Suggestions for Authors

1、Although the introduction section explains the research background, it is slightly lengthy. It is suggested to streamline some content that is not directly related to the model and focus on defining the problem and addressing the shortcomings of existing methods.

2、The details of the method section are very complete, but there are too many descriptions of formulas and modules, which readers may feel a bit "cluttered". It is possible to consider merging explanations appropriately to highlight innovative points.
3、The results display is quite rich, but the chart numbering and explanation sections are somewhat lengthy. It is recommended to emphasize key results and trends in the main text and include some secondary indicators in the appendix.
4、The innovation and contribution points of the discussion section can be further highlighted, such as why multi task learning is better than single task learning, and where the practical application value lies, which can be further strengthened.

5、The legend and coordinate axis explanations of some figures (such as Figure 9 and Figure 10) are not clear enough. It is recommended to label them more clearly for readers to quickly understand.
6、Although compared with 11 baseline models, the lack of comparison with the latest Transformer variants such as ViT or SpectralFormer may be questioned for lack of representativeness.

7、Specific values have been given for parameters such as learning rate and batch size, but there is no explanation of the selection criteria. It is recommended to supplement sensitivity analysis or provide reference sources.
8、There is a significant difference in the distribution of ICRAF and LUCAS data, and although there is clustering division in the article, there is a lack of discussion on whether class imbalance affects the results.
9、The article mentions that prior knowledge soft constraints have a negative impact on some models, but the explanation for the reasons is relatively vague. It is recommended to provide more in-depth analysis or experimental support.
10、The details of the code and model parameters have not been made public. Although the dataset is open, it is best to supplement the open source code or pseudocode process to enhance persuasiveness.

Reviewer 2 Report

Comments and Suggestions for Authors
  1. In the introduction (lines 97–99), the authors present “introducing soft constraints to guide models toward greater robustness” as one of the paper's four major contributions. However, in the abstract (lines 18-19) and the discussion section of the results (lines 396-406), the paper's experimental conclusions indicate that this soft constraint may “amplify intrinsic noise” and even negatively impact models with poorer performance. This creates a clear contradiction between the contribution stated in the introduction and the subsequent experimental findings.
  2. The authors’ proposed soft constraint of “clay + silt + sand = 100%” has limited performance improvement in some models and may even slightly degrade performance in relatively weaker models. The paper’s explanation for this phenomenon is relatively brief, presenting only the average performance of three tasks. The authors are advised to provide a further explanation of the specific performance differences of this constraint across tasks to understand its mechanism better. (Lines 396-406)
  3. The ICRAF and LUCAS datasets cover a wide range of data and are highly representative. The proposed method performs well in these laboratory data scenarios. Furthermore, the paper notes the method’s potential value in large-scale applications. However, the discussion section does not fully elaborate on its potential and limitations for field or remote sensing-scale applications. The authors are advised to provide appropriate additional information in the discussion section to enhance the paper’s applicability and generalizability. (Lines 34-40)
  4. Regarding the research background and application of multi-task learning in soil hyperspectral analysis, the citations of related work are relatively limited. It is recommended that the authors appropriately supplement the references, results, and conclusions of similar studies, such as the application of multi-task learning in soil nutrient prediction, and the prevalence of inter-task conflicts or negative transfer phenomena, to enrich the research background. (Lines 75-85)
  5. The references to deep learning are mainly from the past five years. It is recommended to supplement with more recent related research, especially the latest developments, to ensure the timeliness and comprehensiveness of the research context.
  6. In Section 3.2.1, the paper observes that traditional machine learning models perform poorly (RDP < 2.0). It is recommended to briefly supplement this analysis here, for example, by noting that these models struggle to automatically learn complex feature representations from high-dimensional, nonlinear spectral data. This would better highlight the advantages of deep learning approaches.

Reviewer 3 Report

Comments and Suggestions for Authors

The presented manuscript is of significant practical importance for researchers studying the physical properties of soils.

One advantage of the work is the use of real research data from a significant area.

All illustrations are well-done.

The Materials and Methods section is clearly described.

The Results section is also well-described.

However, the Discussion section is rather short and could benefit from more specific conclusions.

It would be helpful to see the conclusions expanded upon in greater detail.

Reviewer 4 Report

Comments and Suggestions for Authors

The paper aims to design, implement, and evaluate MSRA-Net, a multi-task learning model equipped with dynamic weighting strategies and prior knowledge-based soft constraints for soil texture prediction from visible-near infrared (Vis-NIR) spectra. The manuscript falls within the scope of the journal, as it combines novel sensing, computational, and machine learning strategies for environmental and agricultural monitoring.

Several aspects of the paper demonstrate thorough technical execution and contextual awareness. The selection and preprocessing of two major benchmark datasets (LUCAS, ICRAF) are conducted with appropriate rigor, including quality checks, noise reduction, and equitable partitioning to mitigate sampling biases. The architecture introduces a modular combination of multi-scale grouped convolution, intra-group efficient channel attention, and a branch routing attention mechanism. The integration of homoscedastic uncertainty, dynamic loss weighting, and a domain-inspired soft constraint on the sum of texture fractions is well justified in light of current multi-task learning literature.

However, several critical points require further attention before the manuscript is ready for publication. The most urgent concerns relate to clarity of technical description, reproducibility, depth of comparative analysis, and interpretability of results, as follows reported.

  1. Elements of the experimental design lack precise definition and could undermine reproducibility. For example, the preprocessing step using the Mahalanobis Distance for outlier filtering is described as “removal of approximately 2.5% of the ICRAF dataset and 5% of the LUCAS dataset,” yet there is no explicit threshold or rationale for the final trimming percentage provided; reproducibility would be improved by including exact thresholds or code references. In the model implementation, the manuscript mentions “the number of nodes in each hidden layer of the MLP is reduced by 18 from the previous layer.” However, it does not specify the starting node count or total number of layers, making it difficult to reconstruct the design. Additionally, crucial hyperparameters, such as the learning rate and specifics of the CosineAnnealingLR scheduler, are presented. However, batch normalization, regularization strategies, and dropout rates are only sporadically reported and are not unified across all modules.

  1. While the ablation studies and comparative experiments are a significant strength, the narrative occasionally prioritizes breadth over depth. In particular, although Table 2 and the results section demonstrate the superior performance of MSRA-MT on benchmark datasets relative to baseline deep learning and machine learning models, the manuscript does not sufficiently interrogate outlier cases or error distribution, nor does it include statistical significance testing beyond standard deviation reporting. This diminishes the impact of claims such as “the MSRA-MT model outperforms its single-task counterparts…on two public different scales datasets.” It would be beneficial to include visual or tabular information about the variance of model performance across folds or a confusion/error matrix to more transparently document cases where the model underperforms.

  1. Some technical details are inconsistently presented, affecting overall clarity. For example, descriptions such as “the final output layer is a fully connected layer with a length of 3, responsible for the final predictions of clay, silt, and sand percentages,” do not clarify whether softmax or another activation is used to enforce the sum-to-100 constraint, or if this is left unconstrained apart from the additional loss penalty. Later in the manuscript, it is noted that “the prior knowledge soft constraint may be detrimental to models with poorer initial performance,” but this is hypothesized rather than empirically validated; this central point—concerning the limits of “hard” versus “soft” constraint approaches—would benefit from a more explicit error analysis and visualization.

  1. Fourth, the broader implications and calibration of the MSRA-MT model are insufficiently discussed. The authors state that “this approach provides a valuable reference for identifying potential degraded areas and formulating differentiated governance measures,” yet they do not elaborate on how such predictions can or should be interpreted by domain experts or local authorities, nor do they present a pathway for integrating this into operational sensing pipelines. It is advisable to report whether model deployment on external datasets or in practical field trials was attempted, or, if not, to delimit the as-yet-untested generalizability of the model clearly.

  1. Several minor issues of presentation and referencing persist. The references are numerous and broadly relevant; however, several in-text citations use shortened forms (for example, “Zhang et al. (2017)”) without a consistent style, and some URLs for datasets are embedded directly in the main text with insufficient explanation. The raw code or scripts used for training and evaluation are reportedly available, although they are not explicitly referenced. The inclusion of a GitHub repository or explicit appendices would significantly enhance the manuscript's usefulness to practitioners. A representative example of areas requiring clarification appears in the comparative experiment: “models with inherently poorer performance such as ResNet34-MT and LSTM-MT exhibited a further decline in performance after incorporating the soft-constraint loss term. It suggests that a soft constraint is not universally beneficial for all models.” This insight is valuable but incomplete; an expansion with visual summary statistics and more focused discussion is warranted.

Based on these observations, the manuscript requires major revisions.

Round 2

Reviewer 1 Report

Comments and Suggestions for Authors

The author has made modifications and can be accepted

Author Response

Thank you for your valuable comments on our work. We all agree that these comments are of great significance to our article. Thank you again for taking the time to review our manuscript. We have clearly marked the supplementary parts in orange to address the revision comments raised by one of the reviewers in the second round.

Reviewer 2 Report

Comments and Suggestions for Authors

The manuscript quality has improved, and the current version meets publication standards. I agree to accept it.

Author Response

(The authors gave the same response as above.)

Reviewer 4 Report

Comments and Suggestions for Authors

The authors have addressed each of the major revision points with specific commentary, added discussion, and new figures or references. However, a few issues still warrant further refinement before final acceptance: In particular, as follows suggested:

  • Some technical descriptions (preprocessing thresholds, full reproducibility, reporting of specific hyperparameters across all modules) could still be more explicit for a fully self-contained methodology section.
  • The discussion on how results and models should be interpreted and utilized by domain experts, as well as plans for practical integration into operational sensing, remains somewhat limited and would benefit from a clearer outlook or specific examples.
  • While visualizations and analysis for error and model variance are strengthened, a quantitative statistical significance test (such as pairwise model comparison, outlier breakdown, or confusion/error matrix for each task) is still missing.

Overall, the paper is now much stronger; however, minor revisions are still required.

Author Response

Please see the attachment. All revisions made in response to your feedback have been clearly marked using orange revision indicators. 
